# Quantitative analysis of auxin sensing in leaf primordia argues against proposed role in regulating leaf dorsoventrality

Neha Bhatia[1], Henrik Åhl[2,3], Henrik Jönsson[2,3,4], Marcus G Heisler[1]*

[1]School of Life and Environmental Sciences, University of Sydney, Sydney, Australia; [2]Sainsbury Laboratory, University of Cambridge, Cambridge, United Kingdom; [3]Department of Applied Mathematics and Theoretical Physics, University of Cambridge, Cambridge, United Kingdom; [4]Department of Astronomy and Theoretical Physics, Lund University, Lund, Sweden

**Abstract** Dorsoventrality in leaves has been shown to depend on the pre-patterned expression of KANADI and HD-ZIPIII genes within the plant shoot apical meristem (SAM). However, it has also been proposed that asymmetric auxin levels within initiating leaves help establish leaf polarity, based in part on observations of the DII auxin sensor. By analyzing and quantifying the expression of the R2D2 auxin sensor, we find that there is no obvious asymmetry in auxin levels during Arabidopsis leaf development. We further show that the mDII control sensor also exhibits an asymmetry in expression in developing leaf primordia early on, while it becomes more symmetric at a later developmental stage as reported previously. Together with other recent findings, our results argue against the importance of auxin asymmetry in establishing leaf polarity.
DOI: https://doi.org/10.7554/eLife.39298.001

*For correspondence:
marcus.heisler@sydney.edu.au

**Competing interests:** The authors declare that no competing interests exist.

## Introduction

The mature leaves of seed plants are usually flat with distinct cell types making up their dorsal (upper) and ventral (lower) tissues, which are derived from the adaxial (side closest to meristem) and abaxial (side farthest from meristem) primordium tissues. A fundamental question in plant development is how this tissue patterning is first specified. Recently, we reported evidence in support of a previously proposed hypothesis that dorsoventral leaf patterning is derived from a pre-pattern present in the shoot apical meristem (SAM), where leaves arise (*Hagemann and Gleissberg, 1996*; *Husbands et al., 2009*; *Kerstetter et al., 2001*; *Koch and Meinhardt, 1994*). We found that transcription factors that promote dorsal and ventral leaf cell types are expressed in the SAM in a pre-pattern that determines the orientation of the leaves that subsequently form (*Caggiano et al., 2017*). Additionally, we found that the plant hormone auxin promotes dorsal cell fate by maintaining expression of the Class III HD-ZIP transcription factor REVOLUTA (REV) and repressing KANADI1 (KAN1) expression in the adaxial cells of organ primordia (*Caggiano et al., 2017*). These findings however contrast with that of another study which concluded that leaf polarity is dependent on an asymmetry in auxin levels within leaf primordia, with relatively low levels of auxin in adaxial cells compared to abaxial cells being critical to maintain dorsal identity (*Qi et al., 2014*). This conclusion was based in part on the finding that exogenous auxin application to tomato leaf primordia resulted in the formation of radialized leaves that appeared ventralized. Also, it was found that an auxin sensor, the DII (*Brunoud et al., 2012*; *Vernoux et al., 2011*) indicates low levels of auxin in adaxial leaf tissues compared to abaxial tissues at leaf initiation (*Qi et al., 2014*). Hence asymmetries in auxin concentrations between the adaxial and abaxial tissues in leaf primordia, as a result of PIN1 mediated auxin transport, were proposed to help establish leaf dorsoventral cell type patterning

(*Qi et al., 2014*), in contrast to the proposed pre-pattern mechanism (*Caggiano et al., 2017*). Building further on this conclusion, a more recent study proposed that low levels of auxin in adaxial primordium tissues are necessary to restrict the expression of the WOX1 and PRS genes to the middle domain, since auxin promotes their expression (*Guan et al., 2017*). Finally, the reported asymmetry in auxin has also been linked to asymmetries in the mechanical properties of leaf tissues and their morphogenesis (*Qi et al., 2017*).

## Results and discussion

In-order to investigate the proposed auxin asymmetry proposal in more detail, we decided to examine the distribution of auxin within initiating leaf primordia using the ratio-metric R2D2 reporter, which acts as a proxy for the cellular sensing of auxin (*Liao et al., 2015*). A high DII/mDII ratio indicates relatively low levels of auxin sensing while a high mDII/DII ratio indicates relatively high levels of auxin sensing (*Liao et al., 2015*). We first focused on the incipient stages when PIN1 convergence patterns are established, by looking at later forming primordia (third, fourth or fifth) in seedlings 4-5DAS (days after stratification). In general, during these early stages DII/mDII ratios were extremely low in cells coinciding with high levels of PIN1-GFP expression while on either side of these cells, ratios were higher but to a similar degree (*Figure 1*). Previously, by examining PIN1-CFP together with *pREV::REV-2xYPet* and *pKAN1::KAN1-2xGFP* in similarly staged primordia we found that REV expression can coincide with PIN1 at later stages when it expands peripherally but that KAN1 expression is invariably complementary to cells with high PIN1 expression (*Caggiano et al., 2017*). Overall our observations therefore indicate that KAN1 expression is associated with low auxin sensing cells while REV expression is associated with different levels of auxin sensing depending on the stage of development. We then examined DII/mDII ratios in the leaves after initiation. For this we imaged the first two leaves when they are approximately 4–5 cells long along their proximo-distal axis (3DAS), as well as a day later as they begin to elongate (6–9 cells along the proximo-distal axis; 4DAS) and again included the PIN1-GFP marker in our analysis to correlate cellular auxin sensing with PIN1 expression and polarity. At 3DAS, the expression patterns of REV and KAN1 are already polar within such primordia, although the FILAMENTOUS FLOWER (FIL) expression domain at this stage of development is still being refined (*Caggiano et al., 2017*). At this stage, PIN1 is polarized towards the distal tip of leaf primordia but has reversed polarity away from the primordia, towards the meristem, in cells adjacent to primordia on the adaxial side. According to the ratio-metric auxin sensor R2D2, DII/mDII ratios varied from moderate to high in adaxial cells of the primordia but also varied similarly in abaxial regions - indicating no obvious asymmetry between the two tissue types (*Figure 2A* to C and M-N; *Figure 2—figure supplements 1* and *2*). A low DII/mDII ratio was only found consistently in more distal regions towards the tip of the primordia and in the provasculature, matching the overall pattern of signal from PIN1-GFP. By carefully monitoring DII/mDII ratios from the distal to proximal end of the initiating leaves, the variability in DII/mDII ratios within adaxial and abaxial tissues was clearly apparent (*Figure 2D–L and O–S*, *Figure 2—figure supplements 1* and *2*). This same overall pattern of signal was found in 28 out of 28 leaves that were examined each for 3DAS and 4DAS (14 seedlings each). Lastly, by inverting the ratio and visualizing the mDII/DII (high ratio intensity corresponding to high cellular auxin sensing), we could further confirm that high levels of auxin sensing were only found in the more distal regions towards the tip of the primordia and within the provasculature, where PIN1-GFP was expressed (*Figure 2—figure supplement 3*).

Although the detection of variable R2D2 ratios in adaxial and abaxial cells already argues against a critical role for the distribution of auxin in patterning leaf dorsoventrality, we decided to further investigate the spatial patterns of R2D2 quantitatively. Using PIN1-GFP marked vasculature as a dividing line to demarcate adaxial and abaxial tissues, we manually cropped adaxial and abaxial confocal volumes from our confocal image stacks of 4DAS old leaves expressing R2D2 using ImageJ (Fiji; https://fiji.sc) (*Figure 3A and B*). We then processed these data using a pipeline that included deconvolution of the image stacks, segmentation of nuclear volumes and calculations of R2D2 ratios averaged over the segmented nuclear volumes (*Figure 3C–E*) (Materials and methods). The aggregated ratios from 20 leaves (10 seedlings) illustrated as separate violin plots for adaxial and abaxial leaf tissues reveals high variability between individual nuclei throughout the leaves and very similar ratio distributions between adaxial and abaxial nuclei (*Figure 3F and G*). According to the data, abaxial tissues have a slightly larger proportion of cells with high DII/mDII ratios (low auxin sensing)

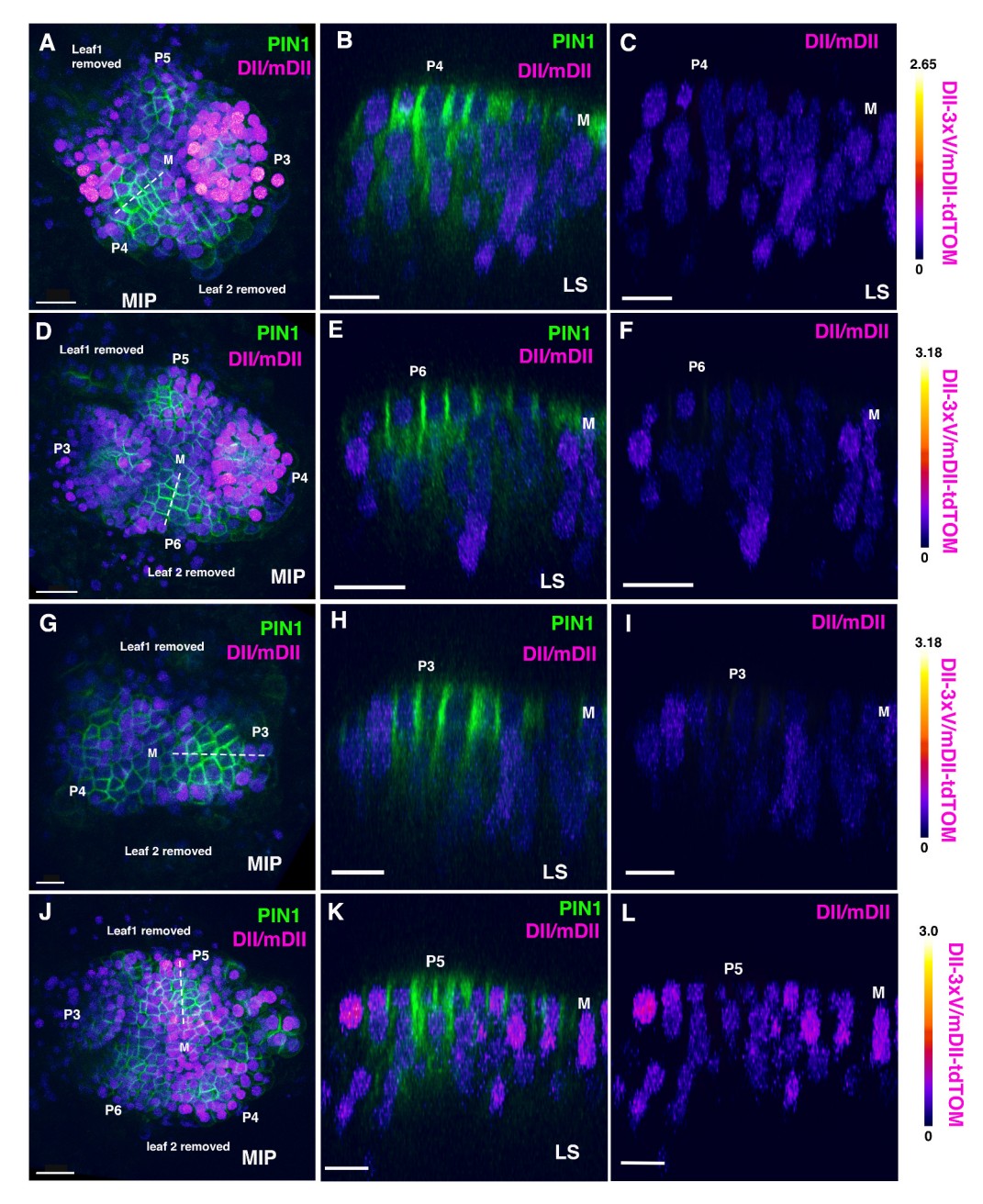

**Figure 1.** Distribution of DII/mDII ratio intensities in incipient leaf primordia (third, fourth or fifth). (A, D, G, J) Confocal projections of *Arabidopsis* seedlings aged 4 or 5 DAS (days after stratification) showing DII/mDII ratio intensity distributions (magenta) along with PIN1-GFP (green) in third, fourth or fifth leaves. (B–C, E–F, H–I, K–L) Corresponding median longitudinal optical sections of incipient primordia along the dashed lines in (A,D,G,J) showing DII/mDII ratio intensity distributions (magenta) along with PIN1-GFP (green) (B, E, H, K) and DII/mDII ratio intensity distributions only (magenta) (C, F, I, L) . Primordia are numbered starting oldest (P3) to youngest (P6). Scale bars 20 μm (B and D), 15 μm (A, C, and F-H) and 10 μm (E). M marks the meristem.

DOI: https://doi.org/10.7554/eLife.39298.002

with a mean of 0.29; standard deviation, SD (0.24); standard error of mean, SEM (0.01) compared to adaxial tissues, which have a mean ratio of 0.24, SD (0.20), SEM (0.004) (*Figure 3G*). Similar distributions are found for individual leaves (*Figure 3F*), although there is leaf to leaf variability, which is also apparent in the spatial distribution of ratios within and between leaves (*Figure 3—figure supplement 1*). Given that the data are not normally distributed we analysed the similarity of the

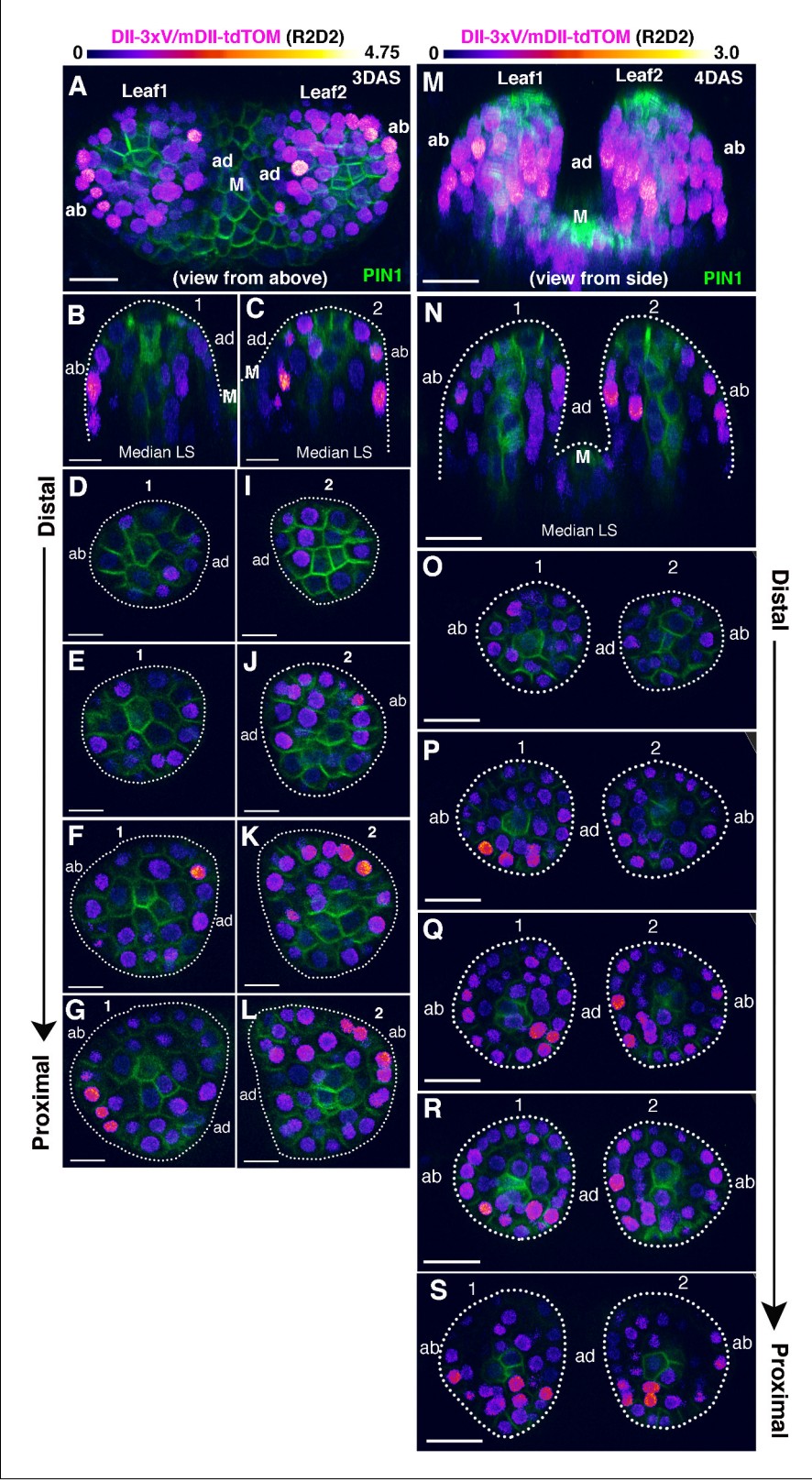

**Figure 2.** Distribution of DII/mDII ratio intensities within initiating leaf primordia (first and second). (**A** and **M**) Confocal projections of *Arabidopsis* seedlings aged 3 and 4 DAS (days after stratification) showing DII/mDII ratio intensity distributions (magenta) along with PIN1-GFP expression (green) in the first two leaves. (**B, C, N**) Corresponding median longitudinal optical sections of first two leaves in (**A**) (**B** and **C**) and in (**M**) (**N**). Note no

*Figure 2 continued on next page*

*Figure 2 continued*

obvious asymmetry in DII/mDII ratio intensities on the adaxial vs abaxial sides of the leaves at 3DAS or 4DAS. (D–L) Optical reconstructions of cross-sections of leaves in (A) along the distal (D) to proximal axis (L). (O–S) Optical reconstructions of cross-sections of leaves in (M) along the distal (O) to proximal axis (S). Note variability in DII/mDII ratio intensities h associated with nuclei within the adaxial and abaxial domains of individual leaves. Scale bars 15 µm (A), 10 µm(B-L), 20 µm (M–S). M marks the meristem.

DOI: https://doi.org/10.7554/eLife.39298.003

The following figure supplements are available for figure 2:

**Figure supplement 1.** Additional examples of 3DAS old seedlings showing Distribution of DII/mDII ratio intensities within initiating leaf primordia (first and second).
DOI: https://doi.org/10.7554/eLife.39298.004

**Figure supplement 2.** Additional examples of 4DAS old seedlings showing the distribution of DII/mDII ratio intensities within initiating leaf primordia (first and second).
DOI: https://doi.org/10.7554/eLife.39298.005

**Figure supplement 3.** Examples of mDII/DII ratio intensity distributions in the first two leaves of 3DAS old Arabidopsis seedlings.
DOI: https://doi.org/10.7554/eLife.39298.006

distributions using a plot of the ranked ratios from aggregated adaxial and abaxial nuclei. The Area Under the Curve (AUC) measure of similarity was found to be 0.556, which again indicates that the distributions are overall very similar (0.5 would indicate they are identical) (*Figure 3H*). All together our results demonstrate that the relative levels of auxin sensing in adaxial and abaxial tissues in young leaf primordia, are overall, very similar. The small difference we identify is that a larger proportion of cells in abaxial tissues display higher DII to mDII ratios compared to adaxial tissues, indicating slightly higher levels of cellular auxin sensing in adaxial tissues, which is the opposite conclusion to that reported earlier (*Guan et al., 2017*; *Qi et al., 2014*). However, we again would like to emphasize that overall, the ratio distributions are very similar.

As our results using the R2D2 auxin sensor indicate a different cellular auxin sensing pattern compared to that reported previously using the DII marker (*Qi et al., 2014*), we decided to re-examined the pattern of DII auxin sensor expression at the same early developmental stages. In contrast to the R2D2 pattern, the DII pattern showed an asymmetry of expression in leaf primordia at 3 DAS, indicating relatively low auxin levels in adaxial primordium cells, as found previously. DII signal appeared strongest in the adaxial epidermis but was also stronger in the adaxial sub-epidermal cell layer compared to abaxial epidermal and sub-epidermal cell layers (*Figure 4- A, C, E and Figure 4—figure supplement 1*) (n = 18/18 leaves, nine seedlings). One day later, DII signal started to show increased relative expression in the abaxial epidermal and sub-epidermal layers compared to earlier stages (n = 14/14 leaves, seven seedlings) (*Figure 4G*). At least for very young primordia, aged 3DAS, our results are similar to those obtained previously and consistent with the proposal that there are low levels of auxin in the adaxial regions of leaf primordia, in contrast to our results using the R2D2 sensor. Given this discrepancy, we next examined the expression of the mDII sensor which is driven by the same 35S promoter as the DII sensor but is not auxin sensitive. Surprisingly we found that, like the DII results, expression of the mDII marker was also higher in the adaxial cells of leaf primordia at 3DAS (n = 16/16 leaves, eight seedlings) (*Figure 4-B,D,F* and *Figure 4—figure supplement 2*). The pattern appeared almost identical to the pattern found using the DII marker except that the mDII marker also exhibited high levels of expression in the shoot meristem whereas the DII sensor did not (compare *Figure 4-A,B*; *Figure 4—figure supplement 1A–G* and *Figure 4—figure supplement 2A–F*). The similarity of expression between DII and mDII was also apparent at 4DAS when both markers started to show expression on the abaxial side as well with no obvious asymmetry in their expression (n = 20/20 leaves, 10 seedlings) (*Figure 4H-Figure 4—figure supplement 3*). To verify the auxin sensitivity of the sensors used we imaged seedlings before and after treatment with 5 mM NAA and found a strong decrease in DII expression compared to mDII and an increase in the mDII-tdTom/DII-V ratio for the R2D2 sensor, consistent with an increase in auxin levels (*Figure 4—figure supplement 4*).

The finding that both the mDII and DII markers, when driven by the 35S promoter, are expressed in an asymmetric pattern during leaf initiation suggests that one reason our conclusions may differ

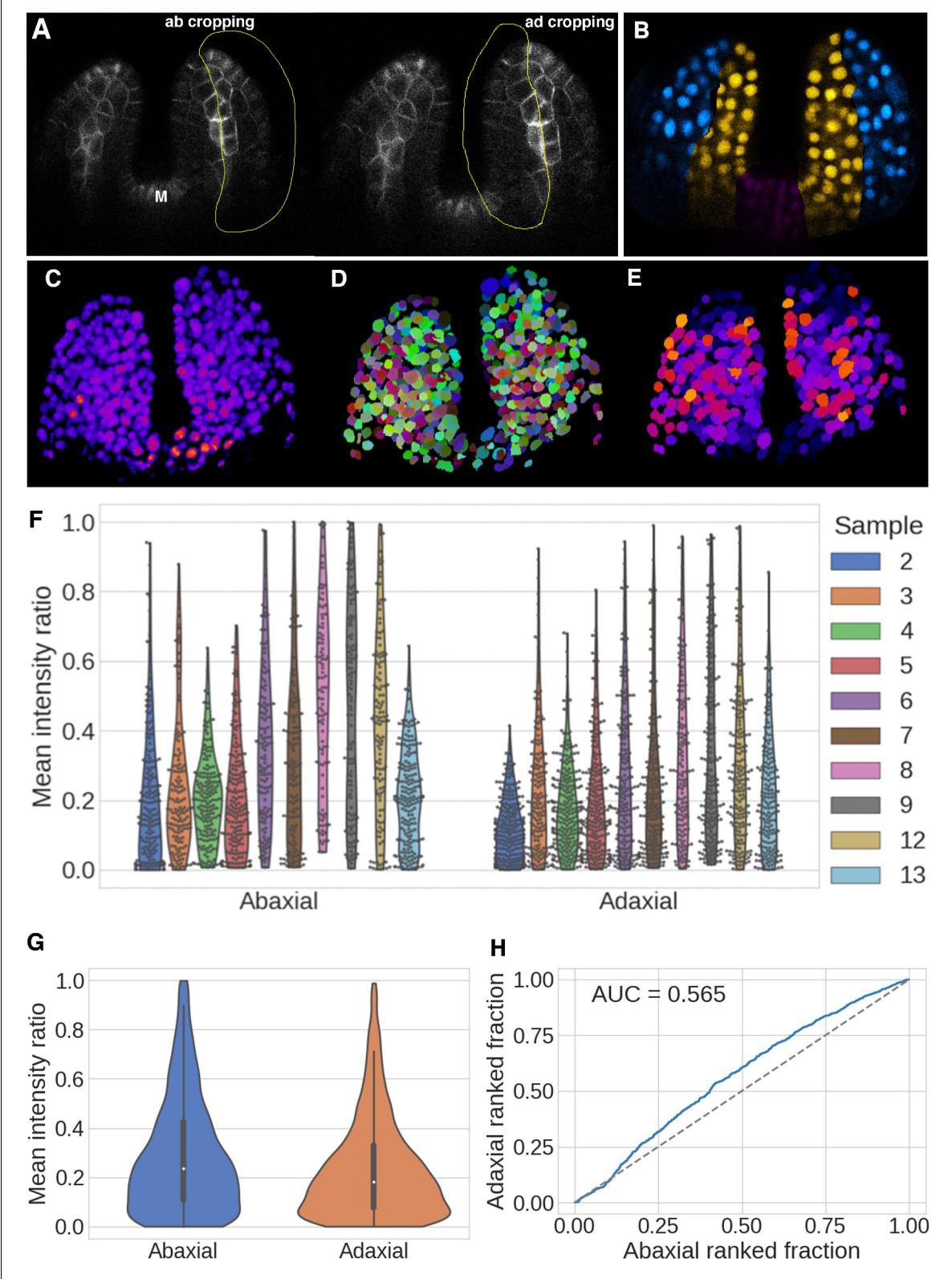

**Figure 3.** Quantification of DII/mDII ratio intensities within the nuclei of the first two leaves in 4DAS old seedlings. (**A**) Representative example of adaxial and abaxial volume estimation and cropping. Snapshot of a single optical slice from a z stack of 4DAS old seedlings showing manual demarcation of adaxial and abaxial cells for cropping (yellow outlines) along the domain of PIN1-GFP expression (gray) in the vasculature. (**B**) Example of resulting nuclear categorisation after cropping, grouped by abaxial (blue), adaxial (yellow), and discarded tissue (magenta). (**C–E**) Quantification

*Figure 3 continued on next page*

*Figure 3 continued*

steps in 3D nuclei, illustrating the initial signal (**C**), subsequent nuclear segmentation (**D**), and the resulting DII/mDII ratio within the nuclear volumes (**E**). (**F and G**) Violin plots of the distributions of the ratios of mean expressions for abaxial and adaxial nuclei after quality filtering, per seedling and all data pooled together. Jitters show the individual data points (**F**), and internal boxplots the median values and distribution quartiles, with whiskers extending to 1.5 times the interquartile range (IQR) (**G**). (**H**) Ranked ratio Area Under the Curve (AUC) plot and score for the distributions. Abaxial n = 1475, adaxial n = 2006, accumulated over 10 seedlings (20 leaves). Distribution values are given as the ratio of normalised mean DII-3xVENUS-N7 over mean mDII- tdTomato (R2D2) expression within the segmented nuclear volumes.

DOI: https://doi.org/10.7554/eLife.39298.007

The following figure supplement is available for figure 3:

**Figure supplement 1.** - Spatial distributions of negative auxin readout Maximal Z-projection of segmented nuclear centroids for 10 seedlings (20 leaves) at 4DAS, coloured by signal ratio after filtering.

DOI: https://doi.org/10.7554/eLife.39298.008

from those reported earlier (*Qi et al., 2014*) is that previously, the DII marker was used to assess relative auxin levels without direct comparison to the mDII marker. We note that although a single section showing expression of the control mDII sensor in older leaves was cited by *Guan et al., 2017* (*Wang et al., 2014*), this information was not adequate for properly assessing similarities and differences, emphasizing the importance of using ratio-metric reporters and quantitative analyses. More recently, these authors have posted a pre-print reporting data for both R2D2 and mDII markers in leaf primordia and concluding that the 35S promoter driven mDII marker is not asymmetric at early leaf stages and that R2D2 ratios again indicate lower auxin levels in adaxial leaf tissues (*Guan et al., 2018*). However for the R2D2 data, the analysis is again based on arbitrarily chosen sections taken from confocal volumes. Regarding the 35S mDII data, the leaves shown have developed to a later stage compared to the leaves we analyze at both 3DAS and 4DAS (4–6 cells long in this study vs 8–9 cells at 2DAS and 6–9 cells in this study vs 9–11 at 3DAS). As our time-lapse imaging shows, the pattern changes rapidly over 12 hr (*Figure 4—figure supplement 3*) and so a staging discrepancy can explain this difference.

In summary, our findings strongly argue against the proposal that asymmetries in the distribution of auxin within young leaf primordia have a strong influence on leaf polarity (*Qi et al., 2014*) or regulate tissue mechanics (*Qi et al., 2017*). Might low levels of auxin in adaxial tissues still limit the pattern *WOX1* or *PRS* gene expression (*Guan et al., 2017*)? In this study, we find high levels of auxin sensing in presumptive provascular cells that also express high levels of PIN1-GFP. Is this where *WOX1* and *PRS* are normally expressed? In fact simultaneous imaging of PIN1-GFP together with PRS-GFP in young leaf primordia indicates that PRS expression is not restricted to cells with high PIN1-GFP expression, although they are both in the middle domain (reconstructed transverse optical section; *Figure 4E* of (*Caggiano et al., 2017*)). More significantly, it was shown that if auxin is applied exogenously, *WOX1* and *PRS* do not become expressed ectopically (*Caggiano et al., 2017*). Overall then, our data argue against a role for auxin in limiting *WOX1* and *PRS* expression to the middle domain. Rather, they support previous results indicating that, like the ventrally expressed KANADI and Auxin Response Factor genes (*ARF2, ARF3* and *ARF4*), the adaxially expressed *HD-ZIPIII* genes restrict auxin response, thereby limiting *WOX1* and *PRS* expression to the middle domain whether the level of auxin in adaxial cells is high or not (*Caggiano et al., 2017*). Our results do however leave other observations unexplained. In particular, to understand the apparent ventralisation of leaf primordia in tomato in response to exogenous auxin (*Qi et al., 2014*) will require further work in assessing how auxin distribution patterns change in response to exogenous treatments and how auxin is distributed during regular development in tomato, preferentially using a ratio-metric auxin sensor such as R2D2.

## Materials and methods

### Plant material and growth conditions

Seeds of the plants expressing *p35S::DII-VENUS* and *p35S::mDII-VENUS* transgenes (*Columbia* ecotype) were obtained from Dr. Teva Vernoux (*Brunoud et al., 2012*). An independent batch of seeds expressing *p35S::mDII-VENUS* transgenes (*Columbia* ecotype) was also obtained from NASC

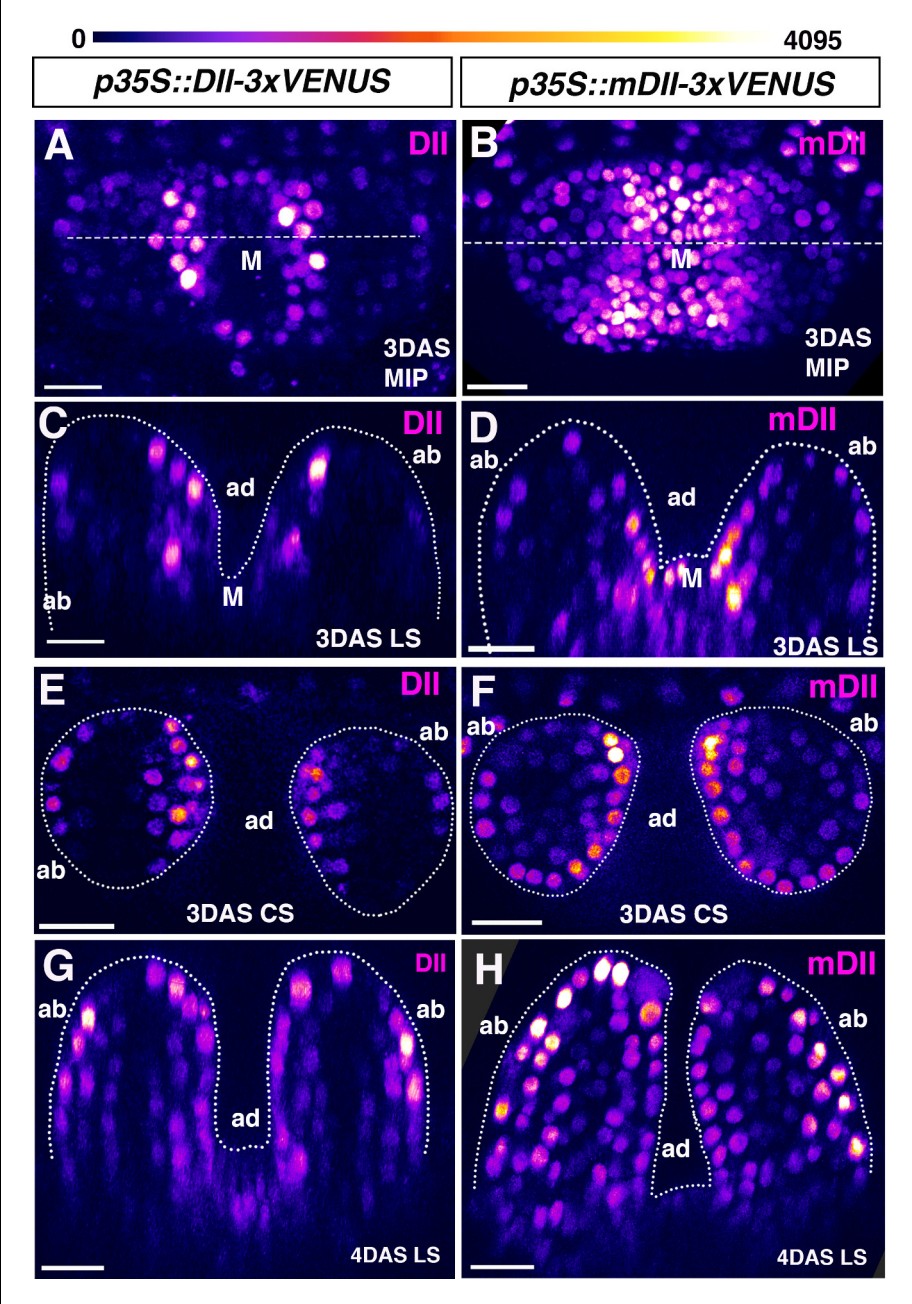

**Figure 4.** Signal intensity distribution in young leaves generated by *p35S* driven auxin sensor and control. (**A and B**) Confocal projections of *Arabidopsis* seedlings aged 3DAS (days after stratification) showing expression pattern of *p35S::DII-VENUS* (magenta) (**A**) and *p35S::mutatedDII-VENUS* (mDII, magenta) (**B**). (**C and D**) Longitudinal reconstructed optical sections of (**A and B**), respectively, along the dashed lines. (**E and F**) Representative examples of transverse reconstructed optical sections of 3DAS Arabidopsis seedlings showing DII-VENUS expression (**E**) and mDII-VENUS expression (**F**). DII-VENUS is more strongly expressed adaxially indicating low auxin sensing on the adaxial side of the leaves relative to the abaxial side. However, mDII-VENUS also shows high expression on the adaxial side of the leaf (compare E with F) and in the shoot meristem. (**G and H**) Representative examples of longitudinal reconstructed optical sections of 4DAS Arabidopsis seedlings showing DII-VENUS expression (**G**) and mDII-VENUS expression (**H**). At this stage, the DII-VENUS shows a more uniform expression and absence of expression in the vasculature. mDII-VENUS also shows a similar pattern to DII but is also expressed in the vasculature (**H**). Scale bars 15 µm (**A and C**) 20 µm B, (**D, E–H**). Figure Supplements.
DOI: https://doi.org/10.7554/eLife.39298.009

The following figure supplements are available for figure 4:

*Figure 4 continued on next page*

*Figure 4 continued*

**Figure supplement 1.** Additional examples *p35S::DII-3xVENUS-N7* expression in 3DAS old *Arabidopsis* seedlings.
DOI: https://doi.org/10.7554/eLife.39298.010

**Figure supplement 2.** Additional examples of *p35S::mDII-3xVENUS-N7* expression in 3DAS old Arabidopsis seedlings.
DOI: https://doi.org/10.7554/eLife.39298.011

**Figure supplement 3.** Changes in *p35S::mDII-V* expression in young leaves over 12 hours.
DOI: https://doi.org/10.7554/eLife.39298.012

**Figure supplement 4.** Response of different auxin sensors to external auxin application.
DOI: https://doi.org/10.7554/eLife.39298.013

(Arabidopsis stock center, NASC ID: N799174) for analysis. R2D2 reporter (*pRPS5::DII-3xVN-7* and *pRPS5::mDII-tdTOMATO-N7*) line carrying *pPIN1::PIN1-GFP* transgene (*Landsberg* ecotype) has been described previously (*Bhatia et al., 2016*; *Liao et al., 2015*). Seeds were germinated and grown on GM medium (pH-7 with 1M KOH) containing 1% sucrose, 1X Murashige and Skoog salts (Sigma M5524), MES 2-(MN-morpholino)- ethane sulfonic acid (Sigma M2933), 0.8% Bacto Agar (Difco), 1% MS vitamins (Sigma M3900) in continuous light.

## Confocal imaging and data analysis

Seedlings aged 3DAS (days after stratification) and 4DAS were dissected as described previously (*Bhatia et al., 2016*). For imaging third, fourth and fifth leaves, the first two leaves of seedlings aged 4DAS, 5DAS or 6DAS were removed using a fine needle (0.45 × 13 mm). For imaging seedlings aged 3DAS, they were positioned and oriented such that the imaging plane was perpendicular to the first two leaves along the dorso-ventral axis. For imaging 4DAS, seedlings were oriented and imaged such that the imaging plane was at 45°−50° angle (by inserting only the tip of the intact cotyledon into the medium) to the first two leaves along the dorso-ventral axis. These orientations were selected to minimize any possible shading and also to re-construct both transverse and optical sections from the same image stacks. Imaging set up of 4DAS used for adaxial-abaxial volume cutting and DII/mDII quantification is described below.

Seedlings were imaged live, on a Leica TCS-SP5 upright confocal laser-scanning microscope with hybrid detectors (HyDs) using a 25X water objective (N.A 0.95). VENUS was imaged using argon laser (excitation wavelength 514 nm) while tdTomato was imaged using a white light laser (excitation wavelength 561 nm). Z-stacks were acquired in a 512 × 512 pixel format, with a resolution of 12-bit depth, section spacing of 1 µm and line averaging 2.

## Seedling-scale ratiometric calculations (*Figures 1* and *2* and associated supplements)

Ratio-metric calculations for R2D2 auxin sensor were performed using ImageJ (FIJI, https://fiji.sc) in two ways.

1. For DII/mDII calculations (negative readout of auxin sensing), DII signal was normalized against mDII (such that $mDII_{max\ intensity} = DII_{max\ intensity}$). A binary mask was generated from mDII channel and was applied to both DII and mDII channel. The resultant DII channel was divided by the resultant mDII channel. Since the values after division were lower than 1, the ratio image was generated as a 32-bit-float and exported as tiff series. PIN1 channel was also exported in 32-bit to visualize the ratio along with PIN1 expression in IMARIS (version 9.2.1) (see below).

2. For mDII/DII calculations (positive readout of auxin sensing), The DII signal was normalized against the mDII signal (such that $mDII_{max\ intensity} = DII_{max\ intensity}$). The mDII-tdTOM channel was divided by the DII-3XV channel. The calculated image was duplicated and then segmented using intensity threshold- based segmentation to create a binary mask with pixel value inside the nuclei one and outside the nuclei to zero. Masking was done to set all the values outside the nuclei uniformly zero. This mask was then multiplied to the original ratio calculated image to set the intensity values in cells with low auxin close to background (zero). The result appeared in a new window and was exported as a tiff series (*Bhatia et al., 2016*).

Both of these protocols are specific for images captured with a resolution of 12-bit.

The tiff series were then opened in IMARIS 9.2.1 (bit-plane) and processed further. Optical sections (transverse or longitudinal) were reconstructed using oblique slicer in IMARIS. If the orientation of the first two leaves was different to each other with respect to the meristem, different oblique slicers were used. This was done to position the slicers parallel to the leaf proximo-distal axis and perpendicular to the dorso-ventral axis to generate precise optical sections. The optical sections are 2 μm in thickness.

## Adaxial-abaxial volume cropping and nuclear quantification (*Figure 3*)

### Data acquisition and pre-processing

DII/mDII ratio quantifications were performed on the leaves of seedlings aged 4DAS. During imaging, seedlings were positioned and oriented such that the imaging plane was perpendicular to the first two leaves along the medio-lateral axis. This orientation allowed a clear demarcation between adaxial and abaxial side based on PIN1-GFP expression in the vasculature. 3D volumes were cropped in Fiji along PIN1-GFP expressing cells in the vasculature as shown in Figure (3A and B). Median optical slice for each leaf was chosen to draw the Region Of Interest (ROI) to crop adaxial or abaxial volumes of the leaves. Before cropping, all the slices were visualized to ensure the chosen ROI covered all the cells in adaxial or abaxial side.

We did not consider potentially cropped (partial) nuclei differently than others during the segmentation. All cropped images were deconvolved using the *psf* Python library (https://www.lfd.uci.edu/~gohlke/code/psf.py.html) for generating point-spread functions, and the Richardson-Lucy iterative algorithm (*Lucy, 1974*; *Richardson, 1972*) in the *Scikit-Image* (v. 0.14) library (*van der Walt et al., 2014*) for restoration. Similarly, we applied a Wiener filter (*Wiener, 1949*) from the Scipy library (v. 1.1.0) (*Jones et al., 2001*). All microscopy settings used for the deconvolution are made available in the supporting code repository.

In order to optimise information for segmenting nuclei, the nuclear signals were merged together so that the intensity value $I_j$ of each voxel $j$ was set to the maximum intensity of the nuclear channels at that voxel

Before segmentation, the images were thresholded using Otsu's method (*Otsu, 1979*). Median and Gaussian filters were then applied sequentially at various iterations to reduce salt-and-pepper noise, and for general smoothing of the input signal.

### Nuclear identification

In order to quantify the input data, we developed a Python implementation of the *ImageJ* plugin *Costanza* (http://www.plant-image-analysis.org/software/costanza), denoted *pyCostanza*, which was further improved for maintainability and flexibility. Both the *ImageJ* plugin and *pyCostanza* are freely available via the Sainsbury Laboratory gitlab repository (https://gitlab.com/slcu/teamHJ/Costanza; copy archived at https://github.com/elifesciences-publications/slcu-teamHJ-Costanza) (*Åhl et al., 2018*).

*pyCostanza* performs marker-less object-identification by identifying gradient attractors in the input image. That is, for every voxel within a given (optional) mask of the image, the algorithm identifies the neighbouring voxel that maximises the value of

$$\Delta I/\Delta x = (I_{neigh} - I_{current})/(x_{neigh} - x_{current})$$

where x denotes the spatial position (unit length) of the current of neighbouring voxel, and $I$ the corresponding intensity. The neighbourhood can be chosen as desired, with common and useful options being spherical, cubical, and cross-shaped neighbourhoods. After identifying the neighbouring voxel maximising the metric, the pair are connected in a graph. Subsequently, the resulting connected regions (basin-of-attractors) after all voxels have been visited correspond to initial approximations of the objects in the image. An attractor is defined as the voxel with the highest intensity within the corresponding basin-of-attractor.

### Object improvement

The initial segmentation is improved upon by a number of merging or removal algorithms, functioning as follows:

- *Merge, Distance*: Attractors within a given (Euclidean) distance of each other are merged together.

- *Merge, Depth*: Neighbouring domains are merged together if the intensity depth of a domain is less than a given threshold. The intensity depth is defined as the difference between the maximal intensity value in either region, and the minimal intensity value in the boundary between the domains.
- *Merge, Small2Closest*: Domains under a certain size (volume) are merged with the closest domain of a volume bigger than the given threshold. A maximal upper limit for whether domains shall be merged or not can be set.
- *Remove, Size*: Domains are removed based on a volume threshold.
- *Remove, Intensity*: Domains are removed based on the (mean) intensity within the domain.

*pyCostanza* also includes functions for performing morphological erosion, dilation, opening and closing operations on the labels.

## Nuclear signal extraction and analysis

The unprocessed DII and mDII signals were normalised under the assumption that the maximal DII signal corresponds to the maximal mDII signal, due to mDII representing non-degrading DII. That is, the DII signal was linearly transformed such that max(DII):=max(mDII) for voxels with a non-background label, resulting in ratios within [0, 1]. The corresponding data per labelled object were extracted from the image using the *Scikit-Image* function *regionprops*. Specific attributes extracted were label centroids, mean DII intensity, and the mean mDII intensity. The intensity ratio, taken as the DII signal relative to the mDII signal, was calculated from these values. For identification of spatial distribution patterns, label centroid coordinates were downprojected in the XY-plane of the input image such that the intensity value of the voxels correspond to the signal ratio (*Figure 3—figure supplement 1*). All nuclei were filtered based on signal strength and size to account for over and under-segmentation, as well as possible included parts of the background.

## Auxin treatment

Seedlings aged 3DAS were dissected imaged and treated with approximately 10 µL of 5 mM NAA (1-Napthaleneacetic acid) solution in water (0.5M stock in 1M KOH) for 60 min and imaged again with same settings as prior to treatment.

## Data availability

Source data files used for all the figures (*Figure 3*) are available via the BioStudies database https://www.ebi.ac.uk/biostudies/studies/S-BSST223). All scripts and software for nuclear segmentation and signal quantification are available via the Sainsbury Laboratory gitlab server (https://gitlab.com/slcu/teamHJ/publications/bhatia_et_al_2019; copy archived at https://github.com/elifesciences-publications/Bhatia_et_al_2019) (*Bhatia et al., 2018*).

## Acknowledgements

We thank Mary Byrne for critical feedback on the manuscript and Teva Vernoux for kindly providing the seeds of plants expressing *p35S::DII-VENUS* and *p35S::mDII-VENUS* transgenes. We also thank Jim Haseloff for providing the seeds of plants expressing *p35S::H2B-mRFP1* and *p35S::EGFP-LTI6b* transgenes. We thank Sevi Durdu for providing technical advice on DII/mDII ratio calculation shown in *Figures 1* and *2* and associated supplements.

## Additional information

### Funding

| Funder | Grant reference number | Author |
|---|---|---|
| Gatsby Charitable Foundation | GAT3395-PR4 | Henrik Jönsson |
| University of Sydney | | Marcus G Heisler |

The funders had no role in study design, data collection and interpretation, or the decision to submit the work for publication.

## Author contributions
Neha Bhatia, Conceptualization, Investigation, Visualization, Methodology, Writing—original draft, Project administration, Writing—review and editing; Henrik Åhl, Data curation, Software, Formal analysis, Validation, Investigation, Visualization, Methodology; Henrik Jönsson, Supervision, Funding acquisition, Methodology, Writing—review and editing; Marcus G Heisler, Conceptualization, Supervision, Funding acquisition, Visualization, Methodology, Writing—original draft, Project administration, Writing—review and editing

## Author ORCIDs
Neha Bhatia (iD) http://orcid.org/0000-0002-2165-5183
Henrik Åhl (iD) http://orcid.org/0000-0002-0655-806X
Henrik Jönsson (iD) http://orcid.org/0000-0003-2340-588X
Marcus G Heisler (iD) http://orcid.org/0000-0001-5644-8398

## Decision letter and Author response
Decision letter https://doi.org/10.7554/eLife.39298.016
Author response https://doi.org/10.7554/eLife.39298.017

# Additional files
## Supplementary files
• Transparent reporting form
DOI: https://doi.org/10.7554/eLife.39298.014

## Data availability
Source data files used for all figures are available via the BioStudies database (https://www.ebi.ac.uk/biostudies/studies/S-BSST223). All scripts and software for nuclear segmentation and signal quantification are available via the Sainsbury Laboratory gitlab server (https://gitlab.com/slcu/teamHJ/publications/bhatia_et_al_2019).

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
