## [Decision Letter]

Thank you for submitting your article "The establishment of dorsoventrality in leaves does not depend on an asymmetry in auxin distribution" for consideration by *eLife*. Your article has been reviewed by Christian Hardtke as the Senior Editor, Dominique Bergmann as the Reviewing Editor and two reviewers. The reviewers have opted to remain anonymous.

This has been a complicated paper to review (as you likely know) and the reviewers and I had a number of discussions about how to proceed. The consensus was that there were two, not totally integrated stories here, one on KAN/REV and a second on auxin distribution. The "Advance" format is really designed to build on a recent *eLife* paper, but the KAN/REV story was not thought to be advanced enough to be suitable for this. On the other hand, the auxin distribution section was a more developed story. Our conclusion is to reroute this into the *eLife* "short report" format, and so we ask that you submit a revision addressing the reviewers' concerns in that form.

A revision focused on the auxin distribution story needs to address the following points:

1) More precise, quantitative analysis on a large number of samples is needed. Specifically,

- Need essential information on the number of plants that were imaged in each experiment at each time point, how many plants showed the patterns shown in the figures, how many plants were discarded, because they suffered growth arrest etc.

- You make the point that the 35S promoter may be highly variable in different transgenic lines and provide two examples with two very different reporter proteins in Figure 4. Since the reporters inevitably will have a massive influence on protein accumulation and hence the measured entity, this experiment does not support the claim made by the authors. It is essential to analyze multiple T1 lines of a single reporter to assess the variability of the promoter without confounding effects by the reporter. Ideally compare variation between 35S and other commonly used promoters, such as Ubiquitin10 or RPS5.

2) The methods need to be described in more detail. Specifically,

- The critical experiments on auxin signaling are poorly described. One example is found in the Results and Discussion section "We first focused on incipient stages where PIN1 convergence pattern is obvious but the expression patterns of REV and KAN are being defined, by looking at later forming primordia (third, fourth or fifth). We found no asymmetries in auxin levels in the future adaxial and abaxial cells (Figure 2).

- I´d like to point out that the reporters used do not measure auxin levels but are proxies for the cellular sensing of auxin. This needs to be corrected throughout the manuscript.

- The color coding of Figure 2 is not explained and simply labeled R2D2 ratio. Does signal suggest absence of auxin sensing or the converse?

- Again, it is quite difficult to know what one is looking at – where is the centre of the SAM, where are the first visible organs?

- You claim that there is no asymmetry in R2D2 signal at the early primordia stage and provide longitudinal optical sections to support this claim. However, depending on which cells you define as being the primordium, there are substantial differences in signal intensities. Most notably, the "outermost cell" that is PIN1-GFP positive consistently appears to have much less R2D2 signal (see Figure 2 E, F, H). You need to be much more careful in interpreting, presenting and describing their data.

- What does "mDII/DII (auxin)" in Figure 3 mean? Is it different from "mDII/DII (R2D2)" in Figure 3—figure supplement 1, or Figure 3—figure supplement 2 (R2D2 ratio), or Figure 3—figure supplement 5 (R2D2 auxin sensor), or Figure 2 (R2D2 ratio)?

3) The discussion of discrepancies between this report, previous reports (and now the new bioRxiv report from Jiao) need a more nuanced discussion.

- While you focus here on auxin distribution mainly, Guan et al. describe a much more complex scenario, where auxin concentrations, the level of several ARFs and the final auxin signaling output level (DR5) do not clearly overlap. This view is not profoundly challenged by your work.

- You argue that previous work by Guan et al., led to erroneous conclusions, because the DII line used shows an auxin independent, asymmetric expression possibly due to the 35S promoter. Proof for this is that another line expressing the auxin insensitive mDII marker shows the same asymmetry. This is in sharp contrast with Guan et al., (2017) or Wang et al., (2014) who did not find this asymmetry for the mDII line (Wang et al., 2014, Supplementary figure 2 for example).

Are these just different interpretations of the same result (i.e. one concludes that the glass is half empty, the other one that it is half full)? If not, what is the cause of this discrepancy (biased selection of results in one of the articles, differences in growth conditions)? The patterns of the different markers used also seem highly dynamic and might even differ between plants at more less the same stage. Could that play a role? Looking at Figure 3G for example, one might conclude that the level of auxin is slightly higher at the adaxial side than at the abaxial side. The same for supplementary Figure 30. If true there might indeed be a (transient) gradient, in contrast to what is concluded. At a slightly later stage, auxin concentrations are higher in the internal (pro) vascular tissues, which is suggested by both R2D2 and DII (e.g. compare Figure 3G/J and H/K). Guan et al., also find this difference between the adaxial side and internal tissues. This implies that there might be less difference between the two markers (and the two manuscripts) than suggested.

---

## [Author Response]

A revision focused on the auxin distribution story needs to address the following points:1) More precise, quantitative analysis on a large number of samples is needed. Specifically,- Need essential information on the number of plants that were imaged in each experiment at each time point, how many plants showed the patterns shown in the figures, how many plants were discarded, because they suffered growth arrest etc.

We have now performed ratiometric analysis of R2D2 on more seedlings than reported in the earlier version and clearly provide the information on number of plants and the leaves imaged in the main text and figure legends. Plants that suffered growth arrest, or damage while dissecting and imaging were not considered.

- You make the point that the 35S promoter may be highly variable in different transgenic lines and provide two examples with two very different reporter proteins in Figure 4. Since the reporters inevitably will have a massive influence on protein accumulation and hence the measured entity, this experiment does not support the claim made by the authors. It is essential to analyze multiple T1 lines of a single reporter to assess the variability of the promoter without confounding effects by the reporter. Ideally compare variation between 35S and other commonly used promoters, such as Ubiquitin10 or RPS5.

Rather than assessing many new T1 lines, the *eLife* reviewing editor suggested an alternative which is to do time-lapse imaging of one line to demonstrate how the expression pattern changes from asymmetric to uniform. We now include such a time-lapse experiment on 3DAS seedlings expressing *p35S::mDII-V* showing a lower mDII signal intensity on the abaxial side of the leaves at 0h but then increased signal intensity within 12 hours (Figure 4—figure supplement 3).

2) The methods need to be described in more detail. Specifically.- The critical experiments on auxin signaling are poorly described. One example is found in the Results and Discussion section "We first focused on incipient stages where PIN1 convergence pattern is obvious, but the expression patterns of REV and KAN are being defined, by looking at later forming primordia (third, fourth or fifth). We found no asymmetries in auxin levels in the future adaxial and abaxial cells (Figure 2).

We have now improved this text (Results and Discussion section).

- I´d like to point out that the reporters used do not measure auxin levels but are proxies for the cellular sensing of auxin. This needs to be corrected throughout the manuscript.

This has been addressed in the beginning of Results and Discussion section. We have replaced the word ‘auxin levels’ with auxin sensing’ throughout the text.

- The color coding of Figure 2 is not explained and simply labeled R2D2 ratio. Does signal suggest absence of auxin sensing or the converse?

We have now provided the colour stamp indicating dynamic range of the intensity based LUT used in the figures and provided a better description of what the ratios imply (DII/mDII or mDII/DII) in the text (Results and Discussion section).

- Again, it is quite difficult to know what one is looking at – where is the centre of the SAM, where are the first visible organs?

We have improved the annotations in all the figures. For example- we have now labelled the meristem with M in all the figures, labelled the position of first two leaves that were removed to expose leaves 3, 4, 5 (P3, P4, P5) in Figure 1 and also indicated where the first two leaves are in Figure 2.

- You claim that there is no asymmetry in R2D2 signal at the early primordia stage and provide longitudinal optical sections to support this claim. However, depending on which cells you define as being the primordium, there are substantial differences in signal intensities. Most notably, the "outermost cell" that is PIN1-GFP positive consistently appears to have much less R2D2 signal (see Figure 2 E, F, H). You need to be much more careful in interpreting, presenting and describing their data.

A major addition to this version of the manuscript is the quantitative comparison of DII/mDII ratio intensities in the adaxial and abaxial halves of young leaf primordia (Figure 3). We did this because we found that DII/mDII ratios appear highly variable and that depending on the optical section examined, different patterns could be perceived from visual examination (Figure 2 and associated supplements). Hence, we used the PIN1-GFP marker as a dividing line to split R2D2 imaging data from whole leaf primordia into adaxial and abaxial halves. We then segmented the mDII signal to demarcate nuclei, calculated the DII/mDII ratios within these nuclei and averaged these results for each nucleus. This analysis thus encompasses all nuclei detected throughout the 3D volume of each leaf. Our results are provided in Figure 3 and clearly show that there is little difference in the distribution of DII/mDII ratio intensities between the two halves. If anything, we find a slightly larger proportion of abaxial cells with high DII/mDII ratios (low auxin sensing) compared to adaxial cells.

- What does "mDII/DII (auxin)" in Figure 3 mean? Is it different from "mDII/DII (R2D2)" in Figure 3—figure supplement 1, or Figure 3—figure supplement 2 (R2D2 ratio), or Figure 3—figure supplement 5 (R2D2 auxin sensor), or Figure 2 (R2D2 ratio)?

We have now made the descriptions clearer. mDII/DII (inverse plotted ratio of intensities of DII and mDII) is an indicator of cellular auxin perception with high intensities corresponding to high auxin sensing (please see above).

3) The discussion of discrepancies between this report, previous reports (and now the new bioRxiv report from Jiao) need a more nuanced discussion.- While you focus here on auxin distribution mainly, Guan et al. describe a much more complex scenario, where auxin concentrations, the level of several ARFs and the final auxin signaling output level (DR5) do not clearly overlap. This view is not profoundly challenged by your work.

A central message from Guan et al., is that low levels of auxin limit WOX/PRS expression in adaxial tissues while ARF2/3/4 repress these genes abaxially – hence they are expressed in the middle domain. In this study, we show that cellular auxin sensing is not lower in adaxial tissues compared to abaxial tissues – instead they are similar. Nevertheless, it could still be argued that low auxin levels may limit WOX/PRS expression in adaxial tissues, as Guan et al., propose. Are WOX1 and PRS normally expressed only in cells with higher cellular auxin sensing than adaxial cells? In this study, we find high levels of auxin sensing in cells that also express high levels of PIN1-GFP (provascular cells). Is this where WOX1 and PRS are expressed? At least for PRS, the answer is no – data from Caggiano et al., show that although both PIN1 and PRS are expressed in the middle domain, PRS is not restricted to those cells expressing high levels of PIN1-GFP in the centre of the leaf (pro-vasculature). Hence PRS expression does not only occur in cells with obviously higher levels of auxin sensing compared to adaxial cells. Perhaps more convincingly, in Caggiano et al., it was also shown that if auxin is applied exogenously, WOX/PRS do not become expressed ectopically. Overall then, the data in both this study and Caggiano et al., argue against a role for auxin being a limiting factor for defining WOX/PRS expression in the middle domain. Instead, these data are consistent with the hypothesis proposed in Caggiano et al., which is that WOX/PRS expression is actively repressed in adaxial tissues by the HD-ZIPIII proteins. We do not argue against Guan et al. in terms of what happens in abaxial tissues.

These points are now better described in the Discussion section.

- You argue that previous work by Guan et al., led to erroneous conclusions, because the DII line used shows an auxin independent, asymmetric expression possibly due to the 35S promoter. Proof for this is that another line expressing the auxin insensitive mDII marker shows the same asymmetry. This is in sharp contrast with Guan et al., (2017) or Wang et al., (2014) who did not find this asymmetry for the mDII line (Wang et al., 2014, Supplementary figure 2 for example).

Asymmetric mDII expression is only observed in very young leaf primordia at 3DAS. Previous studies such as Guan et al., (2017) or Wang et al., (2014) or even the new preprint from Jiao at BioRxiv do not present mDII data in leaf primordia at this early stage, only older stages. This is discussed in detail in our text, with the conclusion being that the data they report is not sufficient to justify their conclusion or properly compare to our data (Results and Discussion section).

Are these just different interpretations of the same result (i.e. one concludes that the glass is half empty, the other one that it is half full)? If not, what is the cause of this discrepancy (biased selection of results in one of the articles, differences in growth conditions)? The patterns of the different markers used also seem highly dynamic and might even differ between plants at more less the same stage. Could that play a role? Looking at Figure 3G for example, one might conclude that the level of auxin is slightly higher at the adaxial side than at the abaxial side. The same for supplementary Figure 30. If true there might indeed be a (transient) gradient, in contrast to what is concluded. At a slightly later stage, auxin concentrations are higher in the internal (pro) vascular tissues, which is suggested by both R2D2 and DII (e.g. compare Figure 3G/J and H/K). Guan et al., also find this difference between the adaxial side and internal tissues. This implies that there might be less difference between the two markers (and the two manuscripts) than suggested.

As the reviewer suggests, a striking feature of R2D2 ratios in young leaf primordia is the variation in pattern. Not only between leaves but even within adaxial and abaxial tissues of the same leaf (Figure 2). Hence being restricted to a limited number of individual sections for analysis easily leads to bias. To eliminate bias in our analysis we therefore undertook a quantitative comparison of DII/mDII ratio intensities in the different sides of the leaf. This involved assessing all detectable nuclei within the adaxial and abaxial tissues of twenty leaves, using PIN1-GFP to mark the boundary between the tissue types. Our quantitative and comprehensive analysis clearly indicates no dramatic differences in R2D2 ratios between adaxial and abaxial nuclei. We conclude that for a case such as this, taking a quantitative and unbiased approach to the cells chosen for analysis is absolutely critical for making a fair conclusion (See Results section).